# Influence of Hydrostatic Pressure and Cationic Type on the Diffusion Behavior of Chloride in Concrete

**DOI:** 10.3390/ma14112851

**Published:** 2021-05-26

**Authors:** Huanqiang Liu, Linhua Jiang

**Affiliations:** 1College of Mechanics and Materials, Hohai University, Nanjing 210098, China; lhjiang@hhu.edu.cn; 2School of Civil Engineering and Transportation, North China University of Water Resources and Electric Power, Zhengzhou 450011, China

**Keywords:** hydrostatic pressure, diffusion source solutions, cation, chloride diffusion coefficient

## Abstract

The durability of the concrete in underground and marine engineering is affected by the underground and ocean environment. Chloride diffusion coefficient under hydrostatic pressure is a key parameter of concrete durability design under corresponding conditions. Therefore, this paper studies the diffusion behavior of chloride in different diffusion source solutions by experiment and simulation. Based on the experimental results, this paper proposes a new chloride diffusion model under the coupling effect of diffusion and convection. The interaction of ions and compounds in the diffusion source solutions, concrete pore fluid, and concrete material are considered in the new chloride diffusion model. The experimental results show that chloride diffusion rate is significantly affected by hydrostatic pressure, which increases with the increase of hydrostatic pressure. The chloride diffusion coefficient shows a certain difference in difference diffusion source solutions. The chloride diffusion coefficient in divalent cationic diffusion source solutions is the largest, the chloride diffusion coefficient in the divalent and monovalent cationic compound ones is in the middle, and the chloride diffusion coefficient in the monovalent cationic ones is the smallest. There is a linear relationship between the chloride diffusion coefficient and the hydrostatic pressure whether in single or combined cationic diffusion source solutions.

## 1. Introduction

Underground and marine engineering are key and hot projects in the engineering field [1,2,3,4,5]. Reinforced concrete in underground and marine engineering are not only corroded by seawater corrosive medium but also subjected to seawater pressure. Seawater and underground water are rich in a variety of ions [6,7], such as Cl^−^, Na^+^, SO_4_^2−^, K^+^, Mg^2+^, Ca^2+^, etc. Chloride has been recognized as the main cause of corrosion of reinforced concrete [8,9,10,11,12,13], and it is an important aspect of the durability research of marine and underground engineering.

Researchers [14,15,16,17,18] have noted that the diffusion behavior of chloride is influenced by the concentration of chloride, the temperature of the environment, and the pore characteristics of concrete, as well as the type of chlorides that compose the diffusion source solution. In most studies or experiments, NaCl the diffusion source solution is used to study the diffusion behavior of chloride. [12,19,20,21,22,23]; however, the components of NaCl diffusion source solution are not similar to those in the actual environment, such as marine engineering and underground engineering environment. With the deepening of research, some researchers [24,25,26] have paid attention to the influence of the cationic type on the diffusion behavior of chloride. Kondo [27] and Ushiyama et al. [28] studied the diffusion behavior of chloride in hardened cement paste under different chloride by diffusion cell and found that the relationship of the chloride diffusion coefficient is D_Cl_ (MgCl_2_) > D_Cl_ (CaCl_2_) > D_Cl_ (LiCl) > D_Cl_ (KCl) > D_Cl_ (NaCl). The study of GjrΦv et al. [29] showed that when the concentration of chloride was the same, the chloride diffusion rate in CaCl_2_ diffusion source solution was larger than that in NaCl diffusion source solution, and he explained the reason by using ions mutual absorption theory. Jiang’s research group [30,31,32,33] studied the diffusion behavior of chloride in concrete in different chloride diffusion source solutions. The results showed that in different chloride solutions, the chloride had a different binding rate with the concrete hydration products during the natural diffusion experiment, and the relationship of the binding rate was: CaCl_2_ > MgCl_2_ > NaCl ≈ KCl. Qiao et al. [34] studied the diffusion behavior of chloride in concrete in a NaCl diffusion source solution and MgCl_2_ diffusion source solution under the coupling effects of diffusion and convection. The results showed that the binding rate of chloride in MgCl_2_ solution was larger than that in NaCl solution. Qiao believed that the difference in chloride binding rate was due to the formation of a new phase Mg(OH)_2_ between MgCl_2_ and concrete hydration products. Zhu [35] carried out natural diffusion tests in NaCl, CaCl_2_, MgCl_2_, and their combined diffusion source solutions. The results showed that the content relationship of free chloride in concrete (*C_f_*) in different diffusion source solutions was as follows: *C_f_*(MgCl_2_) < *C*_f_ (NaCl + MgCl_2_) < *C*_f_ (CaCl_2_ + MgCl_2_) < *C*_f_ (NaCl + CaCl_2_ + MgCl_2_) < *C*_f_ (NaCl + CaCl_2_) < *C*_f_ (NaCl). The content relationship of the total chloride(*C_t_*) was as follows: *C*_t_ (NaCl) < C_t_ (NaCl + MgCl_2_) < C_t_ (CaCl_2_ + NaCl) < C_t_ (NaCl + CaCl_2_ + MgCl_2_) < C_t_(MgCl_2_ + CaCl_2_) < C_t_ (CaCl_2_). Zhu also measured the pH value of the concrete pore fluid, the relationship of the pH values was as follows: MgCl_2_ <MgCl_2_ + CaCl_2_ <CaCl_2_ <NaCl + MgCl_2_ + CaCl_2_ < NaCl. Many researchers [36,37,38] believed that the larger the pH value of concrete pore fluid was, the smaller the amount of chloride Adsorbed by hydration products in concrete.

Concrete pore fluid contains a variety of ions. After chloride and other medium enter the concrete pores, they often have physical or chemical interactions with ions in the pore fluid and concrete hydration products. The interaction among ions will change the diffusion behavior of chloride. Van Quan Tran [39,40] and Guo et al. [41] simulated the interaction among ions based on the numerical model of thermodynamic equilibrium, kinetic control, and surface complexation and confirmed that the interaction among the ions had an influence on the diffusion behavior of chloride. Jyotish Kumar Das et al. [42] studied the diffusion behavior of chloride when different chloride coexist with SO_4_^2−^, and the results showed that the presence of SO_4_^2−^ would reduce the binding rate of chloride, because SO_4_^2−^ was more likely to adsorb on C–S–H.

Jin [43] studied the permeation character of chloride and water in concrete under hydrostatic pressure. The results showed that the penetration depth of water, the penetration depth of chloride and chloride concentration all increased with the hydrostatic pressure, and pressurization time. However, the penetration depth of chloride was only 53% that of water under the same conditions. When the hydrostatic pressure increased from 0 MPa to 1.2 MPa, the chloride diffusion coefficient increased by 500–600%. The chloride diffusion coefficient decreased with increasing of test time under hydrostatic pressure. The hydrostatic pressure made the binding rate of chloride almost zero. TJM Alfatlawi [44] used a self-made hydrostatic pressure device to study the diffusion behavior of chloride in cracked concrete under hydrostatic pressure. The results showed that the penetration depth and concentration of chloride increased with the hydrostatic pressure. Compared with cracks, hydrostatic pressure was the main factor affecting the diffusion behavior of chloride in concrete. It could be concluded that the diffusion behavior of chloride in underwater concrete was mainly affected by hydrostatic pressure. 

The above researchers studied the diffusion behavior of chloride in concrete structures to a certain extent, but the existed research results have some defects, such as the research on the diffusion behavior of chloride in concrete in multi-cationic chloride diffusion source solutions has rarely been reported. The diffusion behavior of chloride in multi-cationic diffusion source solution in concrete under hydrostatic pressure in underground engineering, coastal engineering, bay engineering, and cross-sea engineering is expected to be mastered. In this study, using the self-designed hydrostatic pressure test device, seven kinds of chloride diffusion source solutions composed of single or compound chloride are considered under the hydrostatic pressure of 0.3, 0.5, and 0.7 MPa, respectively. Moreover, in order to more truly reflect the influence of various factors on the diffusion behavior of chloride, a new multifactor coupling model is proposed in this study. The effects of model hydrostatic pressure, the interaction of ions and compounds in the diffusion source solution, concrete pore fluid, and concrete material are considered in this model. The experimental results showed that the multi-factor coupling model can better reflect the diffusion behavior of chloride in underwater engineering. The experimental results of the diffusion behavior of chloride can be better simulated by COMSOL. The simulation results can provide reliable performance parameters for durability design of concrete engineering in similar environment.

## 2. Theoretical Derivation of Diffusion and Convection Coupling Equation

The mechanisms of corrosive medium entering concrete are different under different environmental conditions. In particular, the transport mechanism of ions in the natural soaked concrete is diffusion, and the transport mechanisms of ions in pressurized multi-ionic diffusion source solutions are diffusion and convection. According to Tian [45] and Shao [46], concrete is usually regarded as a continuous homogeneous porous medium to simplify the analysis of corrosive medium transport in concrete; this assumption is also adopted in this study.

Generally, the sum of ions passing through a specific section is called the flux. The flux under the mechanism of diffusion and convection can be expressed as Equation (1):(1)$$J=\sum J_{i}=\sum (D_{i}\nabla c_{i}+c_{i}D_{i}\frac{z_{i}F}{RT}\nabla \varphi +c_{i}v_{i})$$
where *J* is the sum of flux of all types of ions (mol/m^2^·s), *J_i_* is the flux of *i* ions (mol/m^2^·s), *D_i_* is the diffusion coefficient of *i* ions (m^2^/s), Δ*c_i_* is the concentration gradient of *i* ions, *c_i_* is the molar concentration of *i* ions (mol/m^3^), and *v_i_* is the convective velocity of *i* ions (m/s),∇φ is the potential gradient (V), z_i_ is the chemical valence of *i* ions, *F* is Avogadro Constant, *R* is the gas constant, and *T* is the absolute temperature (K). 

To solve the Nernst-Planck equation, an electric neutral balance condition is required. When there is not an external electric field, the current difference between the two ends of the specimen is zero. Referring to Song’s method [25], the Poisson equation can be simplified as Equation (2):(2)$$\sum_{i}^{n}\left(\left|z_{i}^{+}\right|c^{+}-\left|z_{i}^{-}\right|c^{-}\right)=0$$

Under natural diffusion condition, there is no external pressure. However, in the system of multi-ionic diffusion source solutions, the diffusion coefficient of each type of ions is different, and the influence of local electric field should be considered. In this case, the diffusion flux is more reasonable described by Equation (3).
(3)$$\frac{\partial c_{i}}{\partial t}=\frac{\partial }{\partial x}(D_{i}\nabla c_{i}+c_{i}D_{i}\frac{z_{i}F}{RT}\nabla \varphi )$$
where *D_i_* is the diffusion coefficient of *i* ions (m^2^/s), *t* is the diffusion time (s), *x* is the distance from the diffusion source solutions to the intrusion surface (m).

Considering the convection, the convection flux *J_dl_* can be calculated by Equation (4).
(4)$$J_{dl}=\sum J_{idl}=\sum c_{i}v_{i}$$

If the convective velocity and concentration of each type of ions are known, then the convection flux can be calculated by Equation (4). The convective velocity can be calculated using Darcy’s law in conjunction with the continuity equation and the state equation of pore fluid.

Under hydrostatic pressure, the transport mechanism of pore fluid in concrete conform to Darcy’s law, the Darcy velocity of pore fluid can be obtained by Equation (5):(5)$$v=-\frac{k}{\eta }\cdot \frac{\Delta P}{\Delta x}=-\frac{k}{\eta }\nabla P$$
where *v* is the Darcy velocity of pore fluid (m/s), *k* is the hydraulic conductivity (m^2^), *P* is hydrostatic pressure (kg/m/s^2^), Δ*P* is the pressure drop, Δ*x* is the length that the pressure drop is taking place over (*m*), and *η* is the dynamic viscosity coefficient (kg/m/s). Δ*P* is the pressure gradient vector. The negative sign in Equation (5) means that the direction of velocity is opposite to that of the pressure gradient.

According to the hydrostatic pressure equation *P = ρ·g·H*, then the Darcy velocity *v* can be expressed by the water head *H* (m), Equation (5) can be converted to Equation (6).
(6)$$v=-\frac{k\rho g}{\eta }\nabla H$$
where *ρ* the density of the fluid (kg/m^3^), *H* is the water head (m), *g* is the acceleration of gravity (m/s^2^), *η* is the dynamic viscosity coefficient (kg/m/s).

The relationship equation between permeability coefficient (*K* (m/s)) and hydraulic conductivity (*k*) can be expressed as Equation (7):(7)$$K=\frac{k\rho g}{\eta }$$

Substitute Equation (7) into Equation (6), and then the Darcy velocity of pore fluid can be expressed as Equation (8):(8)v=−K∇H

The hydrostatic pressure *P* is used for control during the experiment, for example, the hydrostatic pressure of 0.3 MPa is equivalent to the pressure generated by the 30 m water head, and the value of the acceleration of gravity is taken as 10 m/s^2^ in the calculation.

Where *v* is the Darcy velocity of pore fluid (m/s) in Equation (8). In a one-dimensional penetration, the Darcy velocity equation can be expressed as Equation (9):(9)$$u=-\frac{k}{\eta }\frac{\partial p}{\partial x}=-\frac{k}{\eta }\nabla p=-\frac{K}{\rho g}\nabla p$$
where *u* is the Darcy velocity of pore fluid moving along the axial direction (m/s). It should be noted that this velocity does not represent the actual velocity of fluid, but the Darcy velocity converted from tortuosity during the pore transport process.

According to the continuity equation, Equation (10) can be obtained:(10)$$\frac{\partial }{\partial t}(\rho \varepsilon )+\nabla \cdot (\rho \cdot u)=Q_{m}$$
where ε is the porosity of porous media, *Q_m_* is the mass source term (kg/m^3^·s).

Substituting Equation (9) into Equations (10) and (11) can be obtained:(11)$$\frac{\partial }{\partial t}(\rho \varepsilon )+\nabla \cdot \rho (-\frac{k}{\eta }\nabla p)=Q_{m}$$

After the differential expansion of Equation (11) and defining porosity and density as functions of pressure, according to the chain rule, Equation(12) can be obtained:(12)$$\varepsilon \frac{\partial \rho }{\partial t}+\rho \frac{\partial \varepsilon }{\partial t}=\varepsilon \frac{\partial \rho }{\partial p}\frac{\partial p}{\partial t}+\rho \frac{\partial \varepsilon }{\partial p}\frac{\partial p}{\partial t}$$

According to the definition of compressible fluid, Equation (13) can be obtained:(13)$$\chi_{f}=\frac{1}{\rho }\frac{\partial \rho }{\partial p}$$
where *χ_f_* is the compressibility of fluid. 

Substituting Equation (12) into Equation (13); Equation (14) can be obtained:(14)$$\frac{\partial }{\partial t}(\rho \varepsilon )=\rho (\varepsilon \chi_{f}+\frac{\partial \varepsilon }{\partial p})=\rho S\frac{\partial p}{\partial t}$$
where *S* is the coefficient of storage (1/Pa).

Equation (11) can be expressed as the form of Equation (15):(15)$$\rho S\frac{\partial p}{\partial t}+\nabla \cdot \rho (-\frac{k}{\eta }\Delta p)=Q_{m}$$

The coefficient of storage *S* can be expressed as function of the weighted compressibility of the fluid in pores.
(16)$$S=\varepsilon \chi_{f}+(1-\varepsilon_{p})\chi_{p}$$
where *χ_p_* is the compressibility of concrete. Generally, the compressibility of water is 4.4 × (10^−10^–10^−11^).

Equation (15) applies to the convective system in saturated state; the permeability of the convective system in unsaturated state has to be modified, since we study the transport behavior of chloride in concrete in the saturated state, so Equation (15) is applicable.

Equation (15) is the velocity field obtained by Darcy’s law under the saturated concrete condition.

Then, according to the mass conservation equation:(17)$$\frac{\partial C_{i}}{\partial t}+\nabla \cdot (-D_{i}\nabla C_{i})+u\cdot \nabla C_{i}=R_{i}$$
where *C_i_* is the concentration of *i* ions, *D_i_* is the diffusion coefficient of *i* ions, *R_i_* is the reaction term, and *u* is the Darcy velocity.

In Equation (17), the first term in the left part is the concentration of *i* ions at the moment of *t*, the second term is the diffusion term and self-generated electric field term, and the third term is the convection term. The term in the right part is the reaction term, which is the consumption or production of *i* ions.

For studying the diffusion behavior of chloride in concrete in multi-ionic diffusion source solutions, the adsorption property of chloride must be taken into account [32,47,48], because their adsorption property influence the diffusion behavior of chloride in concrete. The adsorption of chloride in concrete includes physical adsorption and chemical adsorption. Physical adsorption mainly refers to the adsorption of chloride in the C–S–H interlayer, while chemical adsorption mainly refers to the Friedel salt formation by binding chloride and the AFm phase layer. Both physical adsorption and chemical adsorption can cause changes of the diffusion behavior of chloride, during the diffusion of ions, to maintain the electric neutral balance of the system, each ion is affected by itself and its surrounding ions and its diffusion behavior must be affected by the interaction among the ions. Based on this, to study the diffusion behavior of chloride in multi-ionic diffusion source solutions, the general expressions of solution-precipitation kinetics and fluid-solid reaction intrinsic kinetics are used to characterize the interaction among the ions [32].
(18)$$r_{i}=S_{AX}{\left(k_{d}^{AX}-k_{a}^{AX}[A^{+}][X^{-}]\right)}^{n_{i}}$$
where *r_i_* is the reaction rate, and *n_i_* is the reaction order. In this study, all reactions are first-order reactions, so the value is 1; *k _d_ ^AX^* is the dissolution (or desorption) rate constant of *AX* (*AX* is the electrolyte) and, *k_a_^AX^ is* absorption (or formation) rate constants of *AX*, while *S_AX_* is the effective surface area of *AX*. Since *S_AX_* cannot be accurately obtained by calculation, it can be simplified to a relationship proportional to the equivalent concentration.
(19)$$r_{i}=[AX](k_{d}^{AX}-k_{a}^{AX}[A^{+}][X^{-}])$$

The reaction constants of major substances in this experiment are shown in Table 1.

According to the mass conservation equation, the governing equation of ions concentration variation is Equation (19):(20)$$\frac{\partial C_{i}}{\partial t}=-\frac{\partial J_{i}}{\partial x}-r_{i}$$
where *i* refers to Na^+^, K ^+^, Ca^2+^, Cl^−^, OH^−^, and SO_4_^2−^.

The governing equation of solid-phase hydrate content is Equation (20):(21)$$\frac{\partial Cs}{\partial t}=r_{s}$$
where *s* refers to CH, CSH, CSH·CaCl_2_, CSH·2NaCl, CSH·2KCl, CSH·2NaOH, CSH·2KOH, CAH, CASH, and Friedel salt.

## 3. The Establishment of COMSOL Model

According to the theoretical derivation, the interaction among cations strongly influences the diffusion behavior of chloride but how and to what extent is still unknown. The available experimental results [49] of the chloride diffusion coefficient in the single cationic type of diffusion source solutions showed the approximate relation. Song [32,33] proposed an empirical model of the diffusion behavior of chloride in multi-ionic diffusion source solutions but provided no experimental verification. Therefore, it is necessary to establish a corresponding research model to study the diffusion and convection mechanism of chloride in multi-ionic diffusion source solutions.

COMSOL is a professional software for studying the coupling mechanism of multiphysics [50,51], which is developed by Sweden’s COMSOL Inc, Stockholm, Sweden. The dilute mass transfer, and Darcy’s law modules in COMSOL 5.4 are used to simulate the diffusion behavior of chloride in saturated concrete under pressurized diffusion source solutions, boundary conditions, and the network subdivision results are shown in Figure 1. Key parameters of the model are shown in Table 2.

## 4. Materials and Methods

### 4.1. Cement

The cement grade used in this study was Ordinary Portland Cement 42.5 produced by Tianrui Group Co., Ltd., Zhengzhou, China. The main chemical components and technical specifications of the cement are listed in Table 3 and Table 4, respectively.

### 4.2. Aggregate

#### 4.2.1. Fine Aggregate

The fine aggregate used in this study was the machine-made sand from Ling Ai Stone Material Factory in Jiaozuo, China, tested by JGJ52-2006 Standard for the technical requirements and test method of sand and crushed stone for ordinary concrete. The fineness modulus was 2.6, the apparent density was 2.65 g/cm^3^, bulk density was 1.49 g/cm^3^, the void ratio was 43.7%, the clay lump content was 0, the stone powder content was 0.8%, which belonged to medium sand in zone II.

#### 4.2.2. Coarse Aggregate

The coarse aggregate used in this study was the graded broken stone from Huixian Stone Factory in Xinxiang, China, tested by JGJ52-2006 Standard for technical requirements and test method of sand and crushed stone for ordinary concrete. The apparent density was 2.72 g/cm^3^. The bulk density was 1.55 g/cm^3^, the void ratio was 43.0%, the clay lump content was 0, the clay content was 0.4%, and the crushing value was 6.3%, and the needle-like flaky particles content was 5.9%.

### 4.3. Mixing Water

Mixing water used in this study came from municipal tap water of Zhengzhou, China, which met drinking water quality requirements.

### 4.4. Concrete Mix and its Parameters

The water-binder ratio of the concrete was 0.5; the amount of material per cubic meter of concrete was that: 312 kg cement, 746 kg fine aggregate, and 1118 kg coarse aggregate. According to the experiment, the concrete slump value was 50 mm, which had good cohesiveness and water retention. According to SL352-2006 concrete strength test standard, the compressive strength test results of concrete at 28 days was 33.5 MPa, through equivalent calculation of free moisture content, its porosity was 1.84%. Using the trial algorithm, its pore curvature of 0.0227, the simulation results best-fitted the experimental ones, so the pore curvature value was set as 0.0227. Using the calculation method of PU and Taylor [53], the initial concentration values of the main components in the solid phase and pore fluid were respectively set to:

c(Na^+^) = 0.085 mol/m^3^; c(K^+^) = 0.180 mol/m^3^; c(OH^−^) = 0.180 mol/m^3^; c(Ca^2+^) = 0.001 mol/m^3^; c(SO_4_^2−^) = 0.001 mol/m^3^; c(Ca(OH)_2_) = 9.415 mol/m^3^; c(CAH) = 0.580 mol/m^3^; c(CASH) = 0.295 mol/m^3^; c(CSH·2KOH) = 0.058 mol/m^3^; c(CSH·2NaOH) = 0.044 mol/m^3^; c(c/s) = 1.645 mol/m^3^.

Where c represents the concentration of the substances, and the symbol in parentheses represents ionic type or phase.

### 4.5. Experimental Method

According to the concrete mix in Section 4.4, mixed the concrete and formed the cube specimens of 100 × 100 × 100 mm^3^ dimensions by using a concrete vibrating table to compact them. The specimens were placed on level ground for curing at 20 ± 5 ℃ for 24 h, and then the specimens were demolded and sent to a standard curing chamber (at room temperature of 20 ± 2 °C and relative humidity over 95%) for 14 days. After 14 days, the specimens were taken out, and the two end faces of specimens were cut off along the direction of the forming surface with a concrete cutting machine to ensure that the forming surface of the specimens would not affect the experimental results. The middle 50-mm-thick specimens were taken as experimental specimens. The cut specimens were placed in Ca(OH)_2_ saturated solution at 20 ± 2 °C curing for at least 28 days before they were used for the chloride diffusion experiments under hydrostatic pressure.

### 4.6. Hydrostatic Diffusion Experimental Device

In order to effectively simulate the diffusion behavior of chloride under hydrostatic pressure diffusion source solutions, this study designed a set of hydrostatic test device and test programs, which made it possible to avoid some shortcomings of the test device used in previous researchers [54,55]. The schematic diagram and test pattern of the hydrostatic test device designed by this study is shown in Figure 2. The test device has the advantages of low cost, simple operation and coupling experiment with various environmental factors. There is no ring hoop stress in the test process, which overcomes the influence of ring hoop stress in the test method of impermeability instrument on the test results.

### 4.7. Pressurization Procedures and Sampling

The chloride diffusion experiments under hydrostatic pressure were carried out on the cut specimens with a curing period of no less than 28 days. The specimens were placed into the test device shown in Figure 2. According to different diffusion source solutions, the experiments were carried out at hydrostatic pressure of 0.3, 0.5, and 0.7 MPa, respectively. The concentration of chloride in different diffusion source solutions was guaranteed to be 0.5 M (M represent the unit of molar concentration: mol/L, the same below), while the diffusion time under hydrostatic pressure was 24 h. The hydrostatic pressure was closely monitored during the experiment; once the pressure drop exceeded 5% of the preset value, it was supplemented to the set value in time.

After completing the diffusion experiment under hydrostatic pressure, specimens were removed and fixed on the multifunction drill press for powder extraction after drying the source solutions on their surface with paper towels. To ensure the quality of the powder, a super-hard flat bit with a diameter of 12 mm was used for drilling sampling, with no less than six drilling points for each specimen. The location of the drilling point was made sure to avoid coarse aggregate. The drill press footage controlled the sampling depth of each layer (2 mm). The depth of each sampling hole was 20 mm, and a vernier calipers was used to check every two layers to ensure that the depth of the sampling hole was accurate. Three specimens were tested as a group for each type of diffusion source solution, and the sampling method of each group was consistent. Powder samples of each group in the same layer were collected together and sealed in a zip-lock bag for the chloride content titration experiments. All powder samples obtained were labeled with an indication of the layer, cationic type, and pressure to facilitate the statistical processing.

### 4.8. Chloride Titration

The chloride in concrete can be divided into binding chloride and free chloride, it is generally believed that free chloride played a role in the corrosion of reinforcement. In addition, the results of theoretical derivation and simulation process revealed the performance of free chloride concentration. Therefore, the concentration of free chloride is tested. In order to study the influence of hydrostatic pressure on the binding chloride performance for 24h, the acid-soluble chloride content of samples in NaCl diffusion source solution is tested too. According to the test method of chloride in the SL352-2006 Test code for hydralic concrete, the titration in this study was performed using a ZDJ-4A automatic potentiometric titrator produced by Lei Ci Instrument Factory, Shanghai, China. Each sample was titrated for 3 times, and the average value was taken as the test result.

## 5. Results and Discussion

### 5.1. Model Validation and Simulation

Using the dilute material transfer model and Darcy’s law model in the COMSOL software and setting relevant parameters and boundary conditions reasonably, the diffusion character of chloride in pressurized diffusion source solutions in concrete could be accurately simulated. By drawing a cross-section crossing the midpoint along the diffusion direction, the concentration of chloride at each point on the cross-section can be obtained, which is shown in Figure 3. Figure 3 shows the comparison between chloride simulation and test results. In order to verify the feasibility of the simulation of the model, the simulation was carried out on the test data in reference [43] before the formal simulation. Since part of the model parameters in reference [43] could not be obtained, the necessary model parameters were rationally selected according to the basic information of the literature and the basic properties of concrete. The simulation and test results are shown in Figure 3a. The variation trend of the test results is basically consistent with that of the simulation results, indicating that satisfactory results can be obtained as long as the correct model parameters are set when using the model provided in this paper.

According to the model parameters obtained in Section 3, the diffusion behavior of chloride with different cations diffusion sources under hydrostatic pressure is simulated, as shown in Figure 3b–h. As can be seen from Figure 3b–h, the chloride content gradually increases with the hydrostatic pressure applied to the diffusion source solutions, and the greater the hydrostatic pressure, the deeper the chloride diffuse into the concrete. As shown in Figure 3b–h, the simulation results had a high agreement with the experimental ones in all types of diffusion source solutions. The chloride diffusion coefficient in the corresponding diffusion source solutions obtained from the simulation results and the correlation with the test results are shown in Table 5.

Figure 3b shows the simulation result of the diffusion behavior of chloride in the NaCl diffusion source solution under 0 MPa. The simulation result shows that the D_Cl_ is 0.85 × 10^−10^ m^2^/s and the correlation coefficient with the test result was 0.962. Under 0 MPa, the chloride diffusion coefficient is just 18.2% that of 0.3 MPa, 16.0% that of 0.5 MPa, and 14.5% that of 0.7 MPa. It can be seen that hydrostatic pressure is the main driving force of the diffusion behavior of chloride in concrete.

### 5.2. Influence of Hydrostatic Pressure on the Diffusion Behavior of Chloride

In order to study the influence of hydrostatic pressure on chloride binding, acid-soluble and water-soluble chloride were tested on the samples in NaCl diffusion source solution. Figure 4 shows the water-soluble and acid-soluble chloride content test results in 0.5 M NaCl diffusion source solution under hydrostatic pressure of 0, 0.3, 0.5, and 0.7 MPa for 24 h.

Figure 4 shows that the amount of chloride at the same position increased with the increased of hydrostatic pressure. Under 0.3 MPa hydrostatic pressure, the amount of chloride diffused into concrete increased at least 40% in the section of 10 mm away from the concrete surface than that of 0 MPa. Moreover, when the chloride content of 0.02% was taken as the end boundary, the depth of the end boundary under hydrostatic pressure was at least two times that of 0 MPa. The greater the hydrostatic pressure, the greater the depth of the end boundary. It can be seen that hydrostatic pressure can not only accelerates the diffusion rate of chloride but, also, increases the chloride concentration in the pore fluid. The test results are consistent with results of references [43,56,57].

As can be seen from the test results of water-soluble chloride and acid-soluble chloride in Figure 4, they were basically the same. In other words, the adsorption capacity of chloride on the hydration products in concrete is small for 24 h, the reason of almost no adsorption may be related to the short test period. In addition, the pore fluid was pushed by hydrostatic pressure during a short test time, which affected the adsorption performance of pore wall. which is similar to the references [43,56]. However, in reference [58], the binding chloride was found in the hydration products of concrete after a long time of hydrostatic pressure experiment. Therefore, the influence of adsorption effect on the diffusion behavior of chloride should also be considered in the long time of hydrostatic pressure test conditions.

In order to better express the influence of hydrostatic pressure on the diffusion behavior of chloride in different diffusion source solutions, a histogram (Figure 5) was drawn for chloride diffusion coefficient under three hydrostatic pressure levels in Table 5.

It can be seen in Figure 5 that the chloride diffusion coefficient increases with the increase of hydrostatic pressure. Under the same hydrostatic pressure, the chloride diffusion coefficient in the divalent cationic diffusion source solutions is larger than that in the monovalent cationic diffusion source solutions. Under 0.3 MPa hydrostatic pressure, the chloride diffusion coefficient in the divalent cationic diffusion source solutions is 2–7% larger than that in the monovalent cationic diffusion source solutions. While the value is 11.7–16.4% under 0.5 MPa and the value is 22.4–29.4% under 0.7 MPa.

It can be seen from the test results of chloride diffusion coefficient in Figure 5 and Table 5 that under different hydrostatic pressures, the chloride diffusion coefficient in different diffusion source solutions has a certain relationship. The relationship of chloride diffusion coefficient is $D_{\mathrm{Cl}}(X^{2+})>D_{\mathrm{Cl}}(Y^{+}+X^{2+})>D_{\mathrm{Cl}}(Y^{+})$(D_Cl_ represents the chloride diffusion coefficient, X^2+^ represents the CaCl_2_ or MgCl_2_ source solutions, and Y^+^ represents the KCl or NaCl source solutions). The difference of the chloride diffusion coefficient increases with the increase of the hydrostatic pressure. Under 0.3 MPa, the chloride diffusion coefficient in various diffusion source solutions have little difference. However, when the hydrostatic pressure is greater than 0.3 MPa, the chloride diffusion coefficient has a significant increase in all kinds of diffusion source solutions, especially in divalent cationic diffusion source solutions, which may be related to that there is initial starting pressure that drives the pore fluid to flow [56]. When the pressure is lower than 0.3 MPa (less than the initial starting pressure), the main mechanism of chloride entering into the concrete is diffusion, when the hydrostatic pressure is greater than the starting pressure, the main mechanism of chloride entering concrete is convection. Under the convection mechanism, the speed of ions entering the concrete with water is related to the size of ions [25] and the hydrostatic pressure [59].Firstly, the larger the ionic radii, the more difficult it is to flow in a specific pore, and the greater the hydrostatic pressure, the easier it is to flow in a specific pore. Secondly, when the concrete structure is subjected to hydrostatic pressure, the concrete internal microstructure is subjected to hydrostatic pressure, which may cause pore expansion, and even micro-cracks; therefore, the pore fluid can flow into the concrete more easily. The greater the hydrostatic pressure is, the greater the expansion tension of the pores; the more micro-cracks are generated, the more easily the pore fluid enters the pores, and the more ions enter the pores along with the pore fluid.

### 5.3. Influence of the Type of Cations on the Diffusion Behavior of Chloride

Figure 6 shows the simulation results of chloride diffusion source solutions with different cations combinations. It can be seen, in Figure 6, that the types and compositions of the cations in the diffusion source solutions have a certain influence on the diffusion behavior of chloride in concrete. It can be seen in Figure 6 that the chloride diffused faster in the divalent cationic diffusion source solutions, and the concentration of chloride at the surface of the concrete is also higher. It is also shown in Figure 6 that the diffusion behavior of chloride in combined cationic diffusion source solutions is influenced by the cationic types, their valence values, and the corresponding content of cations. For the diffusion source solutions combined by Na^+^ + K^+^ or Ca^2+^ + Mg^2+^, the diffusion behavior of chloride in concrete was similar to the corresponding valence single cationic ones (Figure 6a–c). For the diffusion source solutions combined by Na^+^ + Ca^2+^, the chloride diffusion coefficient is between that of single cationic type of Na^+^ and that of single cationic type of Ca^2+^ diffusion source solution (Figure 6b). The results show that the compositions of the solutions influence the diffusion behavior of chloride. In particular, the valence of cations in the diffusion source solutions has a strong influence on the diffusion behavior of chloride.

The reason for this result may be related to the types of cations in the source solutions. Since the different types of cations, the compositions of pore fluid are different, resulting the different electrostatic field sizes [59] and different ionic radii [56]. Ionic radii of different types of cations [45] is shown in Table 6. Debye-hückel proposed the theory of ion mutual absorption as early as 1923 and analyzed it with the concept of ion atmosphere. The size of ion atmosphere is related to ion strength. The larger the ion strength, the smaller the thickness of ion atmosphere, and the greater the local potential generated, and the greater the electrostatic attraction. When the chloride concentration is the same, the ion strength is larger in the divalent cationic diffusion solutions than that in the monovalent cationic diffusion solutions, so the electrostatic field has a greater influence on the diffusion behavior of chloride in the divalent cationic diffusion solutions.

According to the adjoint theory [60], cations go along with anions to maintain their electric neutral balance during the diffusion process. For cations with larger radii, it is difficult to pass through the specific pores in concrete, while the interaction force between cations and anions inhibits the forward motion of chloride. It can be seen from Table 5 that the chloride diffusion coefficient in single cationic type of Mg^2+^ diffusion source solution is the largest and that, in single cationic type of K^+^ is the smallest. It can be seen from Table 5 and Table 6 that the smaller the radii of cations, the faster the diffusion rate of chloride under hydrostatic pressure.

There are also researchers [61,62,63,64] who believe that in the natural diffusion, electric double layer is the important factor affecting ion diffusion in the pores, because of concrete pore walls are negatively charged, some of cations near pore walls are adsorbed by the pore walls, to maintain neutral, the diffusion rate of chloride reduces. When hydrostatic pressure is applied, the flow of the pore fluid will disturb the pore walls, which changes the adsorption characteristics of the pore walls, thereby affect the diffusion behavior of ions.

It is reported [32,35,55] that with the increase of hydrostatic pressure, the chloride diffusion coefficient is different from the test results under the same conditions, which may be related to the test method. Most researchers [35,55] have studied diffusion behavior of chloride in concrete under hydrostatic pressure using concrete impermeability meter in the experiment, and the schematic diagram of the test method is shown in Figure 7.

In order to avoid the pressure drop in the gap between the specimen and the test mold during the experiment, the sealing material should be used. The mold and the sealing material of penetration experiment will produce a certain ring compressive pressure on the specimen. Under the ring pressure, the specimen shrinks in volume along the ring direction, which results in a shrinkage of the pore diameter in the permeation direction. The shrinkage of the pore diameter leads to a decrease in permeability, which resulting chloride diffusion coefficient decreases accordingly, and the chloride diffusion coefficient obtained is about 10^−12^, which is similar to the results in natural diffusion experiment. The research results of Shao [46] have proved the influence of ring compressive pressure on the diffusion behavior of chloride, and Shao had eliminated the influence of ring compressive by modifying test device in his research. The pressure analysis during the test is shown in Figure 7b. In order to ensure the sealing performance, the sealing pressure is also used, but the direction of the sealing pressure is consistent with that of the hydrostatic pressure. The specimen is compressed in the vertical direction and stretched in the diffusion direction, which increase the pore size in the penetration direction, and the obtained chloride diffusion coefficient is about 10^−10^, which is close to the test result of Shao.

### 5.4. The Relationship Between Hydrostatic Pressure and Chloride Diffusion Coefficient

Figure 8 shows the relationship between hydrostatic pressure and chloride diffusion coefficient. Table 7 shows the fitting equations between hydrostatic pressure and chloride diffusion coefficient in different diffusion source solutions. As shown in Figure 8 and Table 7, there is a linear relationship between chloride diffusion coefficient and hydrostatic pressure applied to the diffusion source solutions. It can be seen in Figure 8 that the slope of the linear model of each single cationic type diffusion solution(K^+^, Na^+^, Ca^2+^, or Mg^2+^) is not consistent, which increases in turn, indicating that the increasing rate of chloride diffusion coefficient is not consistent with the increase of hydrostatic pressure in different cationic diffusion source solutions. The chloride diffusion coefficient is also linearly related to the hydrostatic pressure in the combined cationic diffusion source solutions, and the slopes of the corresponding linear model are related to the type of combined cations. The relationship of the slopes is $a_{Y^{+}}<a_{Y^{+}+X^{2+}}<a_{X^{2+}}$ (a is the slope of the linear model, X^2+^ represents Ca^2+^ or Mg^2+^, Y^+^ represents Na^+^ or K^+^). All the results show that the cationic type in the diffusion source solutions has an influence on the diffusion behavior of chloride in concrete. The fitting equations linking the chloride diffusion coefficient and hydrostatic pressure are listed in Table 7. For a given source solution and the respective hydrostatic pressure, the corresponding chloride diffusion coefficient can be obtained.

## 6. Conclusions

Based on the results and discussion above, the following conclusions can be drawn:The diffusion behavior of chloride in concrete are influenced by the hydrostatic pressure. The diffusion rate of chloride increases with the increase of hydrostatic pressure. Using the professional multi-physics simulation software COMSOL and setting reasonable related parameters, the experimental results can be better simulated, which can provide reliable performance parameters for durability design of concrete engineering in similar environment.The diffusion behavior of chloride is influenced by the cationic type in the diffusion source solutions. The chloride diffusion coefficient in the divalent cationic diffusion source solutions is larger than that in the monovalent cationic diffusion source solutions, and the chloride diffusion coefficient in the small cationic radii diffusion source solutions is larger than that in the big cationic radii ones.The test results have some differences between different test methods. In order to achieve consensus, universal test method needs to be established to study the diffusion behavior of chloride in concrete under hydrostatic pressure.There is a linear relationship (*D_Cl_* = *aP*+*b*) between the chloride diffusion coefficient and the applied hydrostatic pressure in a given diffusion source solution. The determination of parameters in this relationship is related to the cationic type in the diffusion source solutions. Once the parameters *a* and *b* are determined, the chloride diffusion coefficient in this kind of diffusion source solution under different hydrostatic pressures can be obtained.

## Figures and Tables

**Figure 1 materials-14-02851-f001:**
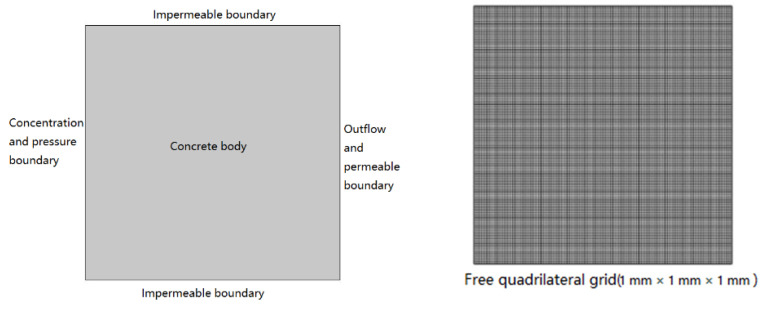
The COMSOL simulation model setup and meshing.

**Figure 2 materials-14-02851-f002:**
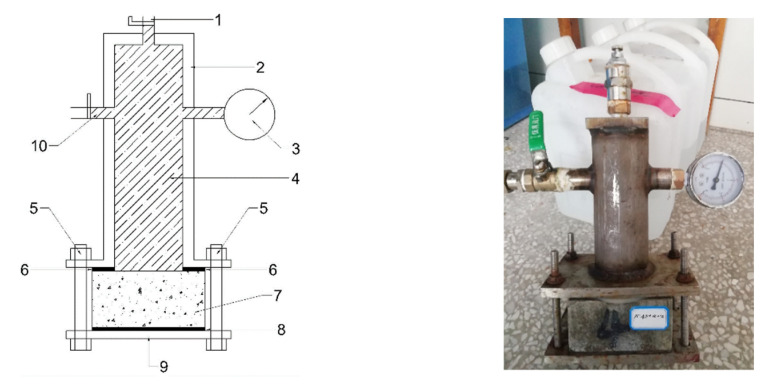
Schematic of the hydrostatic device and test pattern. 1—air evacuation valve; 2—hydrostatic device; 3—hydrostatic pressure indicator; 4—source solution cavity; 5—bolts; 6—hydro elastic rubber ring; 7—experimental specimen; 8—rubber gasket; 9—steel plate; 10—one-way refueling valve(connect the diffusion source solution).

**Figure 3 materials-14-02851-f003:**
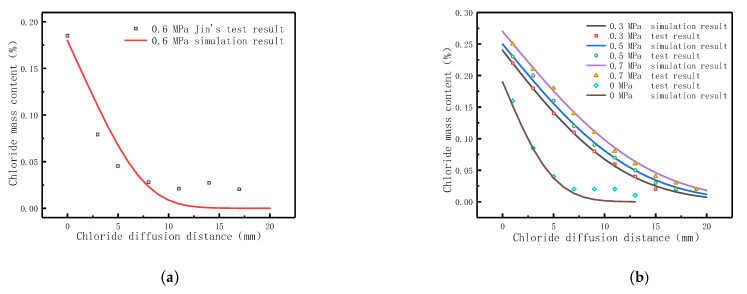
The simulation and test results on diffusion behavior of chloride in pressurized diffusion source solutions. (**a**) 3.5% NaCl diffusion source solution; (**b**) 0.5 M NaCl diffusion source solution; (**c**) 0.5 M KCl diffusion source solution; (**d**) 0.25 M CaCl_2_ diffusion source solution; (**e**) 0.25 M MgCl_2_ diffusion source solution; (**f**) 0.25 M NaCl + 0.25 M KCl diffusion source solution; (**g**) 0.25 M NaCl + 0.125 M CaCl_2_ diffusion source solution; (**h**) 0.125 M MgCl_2_ + 0.125 M CaCl_2_ diffusion source solution.

**Figure 4 materials-14-02851-f004:**
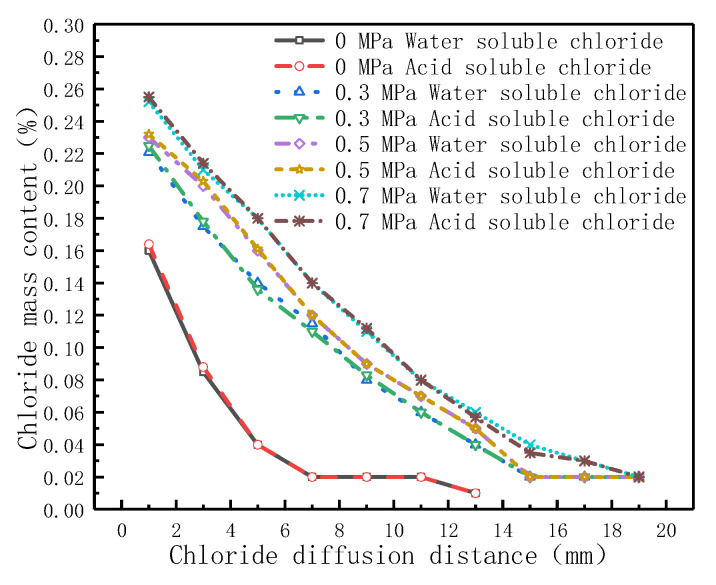
Test results of chloride in NaCl diffusion source solution under different hydrostatic pressures.

**Figure 5 materials-14-02851-f005:**
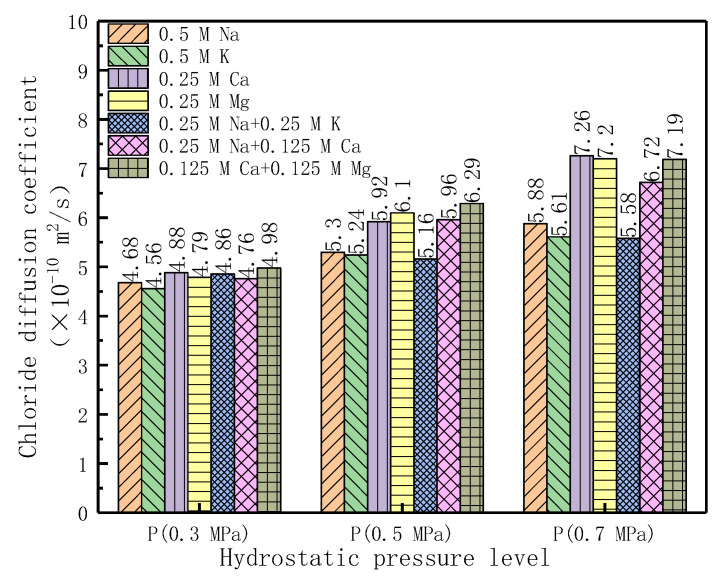
Histogram of chloride diffusion coefficient and the hydrostatic pressure.

**Figure 6 materials-14-02851-f006:**
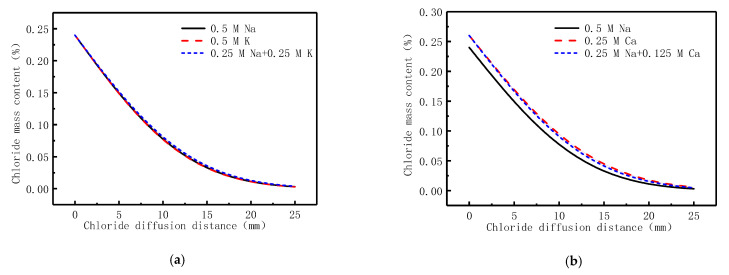
Simulation results of the diffusion behavior of chloride in different diffusion solutions under 0.5 MPa hydrostatic pressure. (**a**) Monovalent cationic diffusion source solutions; (**b**) Monovalent and Divalent cationic diffusion source solutions; (**c**) Divalent cationic diffusion source solutions; (**d**) Combined cationic diffusion source solutions.

**Figure 7 materials-14-02851-f007:**
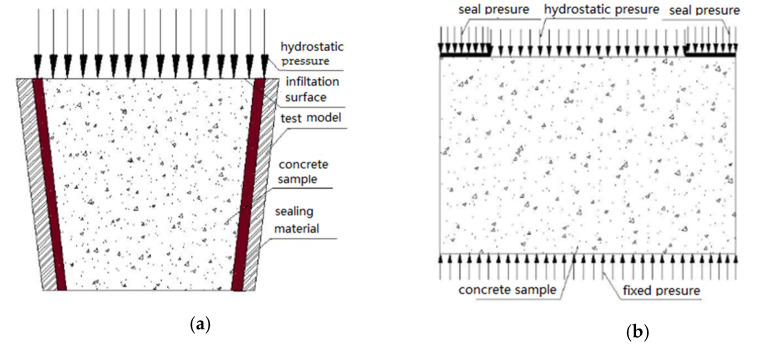
Schematic diagram of the test method of chloride diffusion under hydrostatic pressure. (**a**) Concrete impermeability meter method; (**b**) Test method in this paper.

**Figure 8 materials-14-02851-f008:**
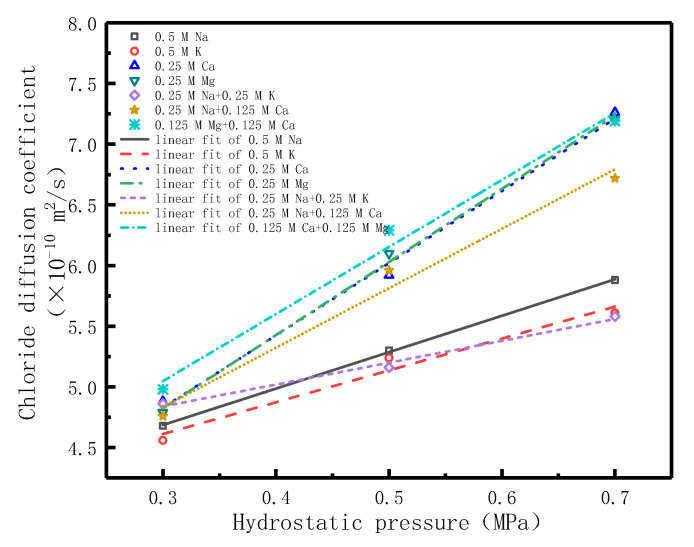
The relationship between chloride diffusion coefficient and hydrostatic pressure.

**Table 1 materials-14-02851-t001:** The reaction constants of the major substances.

Substance	*k_d_* × 10^−7^	*k_a_* × 10^−7^	Substance	*k_d_* × 10^−7^	*k_a_* × 10^−8^
Ca(OH)_2_	0.55	3.70	CSH·2KOH	7.20	0.14
CSH	0.26	0.87	CAH	0.06	2.35
CSH·CaCl _2_	1.80	1.60	CASH	0.05	0.20
CSH·2NaCl	1.80	0.19	CAH·CaCl_2_	2.20	2.16
CSH·2KCl	2.43	0.33	Mg(OH)_2_	—	1.05
CSH·2NaOH	2.14	0.42	CSH·MgCl_2_	0.34	0.48

**Table 2 materials-14-02851-t002:** Key parameters of the model.

Parameter	Expression	Value	Description
Hp	30,50,70 m	30,50,70 m	water head
K	4.06 × 10^−9^ m/s	4.06×10^−9^ m/s	permeability coefficient of saturated concrete
S	ε_p_ × (4.4 × 10^-10^) + (1−ε_p_) × 0	5.28 × 10^−11^	the coefficient of storage of water
D(Ca^2+^)	τ × 0.79 × 10^−9^ m^2^/s	1.7933 × 10^−11^ m²/s	the diffusion coefficient of Ca^2+^
D(Mg^2+^)	τ × 0.7 × 10^−9^ m^2^/s	1.589 × 10^−11^ m²/s	the diffusion coefficient of Mg^2+^
D(Na^+^)	τ × 1.33 × 10^−9^ m^2^/s	3.0191×10^−11^ m²/s	the diffusion coefficient of Na^+^
D(K^+^)	τ × 1.96 × 10^−9^ m^2^/s	4.4492×10^−11^ m²/s	the diffusion coefficient of K^+^
D(SO_4_^2−^)	τ × 1.07 × 10^−9^ m^2^/s	2.4289×10^−11^ m²/s	the diffusion coefficient of SO_4_^2−^
D(OH^−^)	τ × 5.3 × 10^−9^ m^2^/s	1.2031×10^−11^ m²/s	the diffusion coefficient of OH^−^

NOTE: The diffusion coefficient of ions in water is obtained from reference [52]. *τ* is pore curvature, and obtained in Section 4.4.

**Table 3 materials-14-02851-t003:** The main chemical components of the cement (wt.%).

CaO	SiO_2_	Al_2_O_3_	Fe_2_O_3_	MgO	Na_2_o	K_2_O	SO_3_	LOI
59.3	24.7	3.66	3.42	1.51	0.25	0.36	2.56	2.80

**Table 4 materials-14-02851-t004:** The technical specifications of the cement.

Specific Surface Area (m^2^/kg)	Setting Time (min)		Compressive Strength (MPa)		Rupture Strength (MPa)	
	Initial Set	Final Set	3d	28d	3d	28d
347	155	260	31.2	51.2	6.4	8.4

**Table 5 materials-14-02851-t005:** Chloride diffusion coefficient in pressurized diffusion source solutions (0.5 M Cl^−^).

Cationic Type	Chloride Diffusion Coefficient (×10^−10^ m^2^/s)		
	0.3 MPa	0.5 MPa	0.7 MPa
0.5 M Na^+^	4.68 (0.96)	5.30 (0.97)	5.88 (0.99)
0.5 M K^+^	4.56 (0.98)	5.24 (0.98)	5.61 (0.99)
0.25 M Ca^2+^	4.88 (0.96)	5.92 (0.97)	7.26 (0.99)
0.25 M Mg^2+^	4.79 (0.96)	6.10 (0.98)	7.20 (0.98)
0.25 M Na^+^ +0.25 M K^+^	4.86 (0.95)	5.16 (0.92)	5.58 (0.94)
0.25 M Na^+^ + 0.125 M Ca^2+^	4.76 (0.91)	5.96 (0.90)	6.72 (0.91)
0.125 M Mg^2+^ + 0.125 M Ca^2+^	4.98 (0.95)	6.29 (0.97)	7.19 (0.98)

Note: The value in parentheses after the chloride diffusion coefficient is the correlation coefficient: R^2^.

**Table 6 materials-14-02851-t006:** Ionic radii of different types of cations.

Cationic Type	Na^+^	K^+^	Ca^2+^	Mg^2+^
Ionic radius (nm)	0.102	0.138	0.1	0.072

**Table 7 materials-14-02851-t007:** Fitting equations of the chloride diffusion coefficient and hydrostatic pressure.

Cationic Type	Fitting Equation	Correlation Coefficient (R^2^)
0.5 M Na^+^	$D_{\mathrm{Cl}}=3.0p+3.79$	0.999
0.5 M K^+^	$D_{\mathrm{Cl}}=2.63p+3.82$	0.944
0.25 M Ca^2+^	$D_{\mathrm{Cl}}=5.95p+3.04$	0.989
0.25 M Mg^2+^	$D_{\mathrm{Cl}}=6.02p+3.02$	0995
0.25 M Na^+^ +0.25 M K^+^	$D_{\mathrm{Cl}}=1.80p+4.30$	0.982
0.25 M Na^+^ + 0.125 M Ca^2+^	$D_{\mathrm{Cl}}=4.90p+3.36$	0.967
0.125 M Mg^2+^ + 0.125 M Ca^2+^	$D_{\mathrm{Cl}}=5.52p+3.39$	0.977

Note: D_Cl_ is the chloride diffusion coefficient (×10^−10^ m^2^/s); P is the hydrostatic pressure (MPa).

## Data Availability

The data is contained within the article.

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
