# Peer review of "Influence of Hydrostatic Pressure and Cationic Type on the Diffusion Behavior of Chloride in Concrete"

_materials, 2021, doi:10.3390/ma14112851_

Round 1

Reviewer 1 Report

Article deals with influence of cation type on the diffusion of chlorides with and without pressure. Provides valuable experimental data and also an approach for modelling the behaviour. 

Avoid the Dx2 Dy and DY+ terms in the abstract, as it doesnt not mean anything on its own. Better to explain this in words.

Chlorine is gas, Chloride - means one chloride ion, chlorides - means many chloride ions. Please correct throughout the text as appropriate.

Lines 56-58, if Mg reacts with cement hydrates and form Brucite, how does that accelerate the flow of chlorides. Continue the explanation.

Why are you referring to hydrostatic pressure in MPa, wouldnt it be easier to state it as head of water, ie meter or mm? Please give a scale of this in the experimental design section (3.6), so that the calcs are clear.

Line 93 - chloride erosion is a wrong phrase.

Line 104 - delete the word diffusion. Just state it as flux. Also delete the first diffusion word in Lone 105. Rest is ok.

In Eq 5 and Eq. 6, there are no p or H terms. There is change in p and change in h.

velocity in Eq 8 is not explained. It is not clear how you got from Eq. 6 to eq. 8. You have to state that differetiating with respect to time.

Reference all need to be consistent in font, size and style.

Lines 354-360 - smaller cations leads to greater flow. Would you not see this effect even when there is no hydrostatic pressure? Your statements indicate that 0.3 MPa onwards the above behaviour is evident. Does that mean, when there is no pressure, larger cations flow faster? Please elaborate. If it is outside the test remit to comment, please state so.

Figure 3 - does not show any difference between the cation types studied. All of the chloride profiles look comparable (within margin of error). You may want to find a better way to express these results, if there is a significant difference. Quantity of chlorides is one such entity.

Table 6 shows that for 0.3 MPa there is no difference between the various diffusion coefficients. Variation with pressure is however evident.

Similar observation for Figure 4.

Lines 437-440, remove teh word opposite to the radius of cation... the word opposite is causing confusion. Just state that the "which agrees with the size of cations presented in table 7"

All figures, keep the Diffusion Coefficient in m2/s. Please avoid mm2. Refer to Figure 7.

Author Response

Please see the attchment.

Reviewer 2 Report

It is well known that the diffusion coefficients of a liquid in a porous solid used hydrostatic pressure.
In table 8, Best fit equations of chloride diffusion coefficient and hydrostatic pressure for dif- 497 different combinations of cations, fitting equations with just three points, obviously gives good regression coefficients. The authors must be read: "Phillipsite and Al-tobermorite mineral cements produced through low-temperature
water-rock reactions in Roman marine concrete" , American Mineralogist, Volume 102, pages 1435-1450, 2017. In this article reserchers study the Pozzolanic reaction of volcanic ash with hydrated lime is thought to dominate the cementing fabric and durability of 2000-years old Roman harbor concrete. It si a very interesting article to study.

Reviewer 3 Report

The article presents the results of the diffusion behavior of chloride in concrete structures. The following remarks must be considered before paper publication:

  1. The article must be carefully read and corrected - there are many "small" mistakes in the text. In many places, the Authors used capital letters (why?). The description of the symbols in the equations is often incorrect. Exemplary: lines 131-134: "...where k is the liquid permeability(m/s),η is the liquid viscosity(pa·s), p is pressure(pa)". There is "pa" in equation (5). The text must be absolutely read and rewritten.
  2. The results obtained are quite predictable, so please add at the end of the introduction the clear information about the novel elements of the paper.
  3. Line 270:  "For the 28d concrete cube block, the compressive strength was 33.5 MPa" - how many blocks were destroyed during material tests?
  4. Line 275-279 - the sentence is absolutely unreadable.
  5. Fig. 2 - the scale of the figure is disturbed.
  6. Fig. 3 must be corrected. The legend is unreadable. Please add the grid in the charts.
  7. Why Figure 4 is divided into two parts?

Reviewer 4 Report

This paper has been investigated the effect of pressure on the movement of ions in concrete. It is clear that pressure affects the movement of ions. Therefore, it is necessary to consider the following points.

To clarify the effect of pressure, the result at 0 MPa is required.

Also, in recent years, a model that considers the movement of ions in the diffuse double layer in pore has been proposed, but is it not considered in this study?

It is necessary to measure the amount of chloride ions that are fixed as well as the amount of free chloride ions. Furthermore, when referring to cations in the paper, it is necessary to measure the distribution of cations in concrete by the experiment.

Round 2

Reviewer 2 Report

Into marine underground and ocean environment SO42-, K+, Mg2+, Ca2+ levels, are very lower than 0.5 M. For this raison I thinks that it is not interesting study these cation-chloride sistem  influence.

Reviewer 3 Report

Thank yoo for addressing all reviewer's comments.

Reviewer 4 Report

The result of 0 MPa is included only in Fig. 4, and other figures have not been modified. In addition, there is almost no adsorption  in the result of 0 MPa. This is a different result from previous studies, and it seems that there is a problem with the experimental method. Therefore, this paper is  insufficient for publication in the journal.
